# Scientific Stylisation or the 'Democracy Dilemma' of Graphical Abstracts

## Carmen Sancho Guinda

Departamento de Lingüística Aplicada a la Ciencia y la Tecnología, Escuela Técnica Superior de Ingeniería y Diseño Industrial, Universidad Politécnica de Madrid, 28012 Madrid, Spain; carmen.sguinda@upm.es

**Abstract:** The need for more democratic models of interaction between scientists and non-expert audiences, the current commodification of research and the advancements of digital affordances have recently given rise to new online genres for science dissemination, such as the graphical abstract, increasingly demanded by high-impact journals despite its uncertain function. In this paper, I examine the problems posed by this hybrid genre as to the implementation of dialogical and more democratic models of science dissemination; namely, inferential confusion of concepts and narrative sequences, trivialisation and overall interpretive complexity, all of them caused or affected by visual stylisation. After scrutinising over 1000 graphical abstract samples from science blogs, research networks and random finds published in specialised high-rank international journals, I provide a taxonomy of stylisation and make the case for the explicit visual literacy training of students and scholars, as well as for a higher level of specification in the guidelines for potential authors of scientific journals.

**Keywords:** graphical abstracts; genre hybridity; stylisation; interpretive complexity; visual literacy

## 1. Introduction: The 'Democratic Turn' in Science Dissemination. Is It Truly So?

Digital affordances and the pursuit of more democratic paradigms of science dissemination have introduced new discursive practices in academic discourse. Today scholars are encouraged to step down from their 'ivory towers', attract audiences other than experts and communicate with them directly, without the mediation of science journalists. The so-called 'general public' is not so 'general' or homogeneous anymore and may be composed of experts from other fields as much as laypeople. Thus, the idea that the publics are untrained in scientific matters and therefore easily persuadable and in need of content simplifications and popularisations, as the deficit model of science dissemination has traditionally held, does not always apply, and even if it did, more dialogic models of dissemination are underway. Stocklmayer [1] documents the existence of more dialogic and participative models that listen attentively to the citizenship's demands and criticisms, allow the exchange of views between scientists and non-experts, and the joint negotiation of meanings and research agendas. For example, PETA (People for the Ethical Treatment of Animals)[1] is launching an intense online campaign against the experiments conducted on marmosets by scientist Margaret L. Walker at the University of Massachusetts to investigate the absence of menopause among those primates, supposedly subjected to abusive and unnecessary cruel treatments in the laboratory. These facts have been alleged and brought to light by many U.S. pro-animalist activists and groups, whose voices are gaining quite an extensive presence online. Nowadays audiences want to know where their tax money goes, what type of scientific research and projects it is funding and with what results, and to have a say in research policies and courses of action, formerly opaque. They demonstrate actively against research initiatives that they deem pointless or unethical and become ever more visible through associations, ad hoc websites and social media.

Among other recent research genres, typical texts and interaction outcomes of this change in paradigm are science café sessions (usually organised by universities and research

centres to foster close contact with scientists), science blogs of individual or institutional creation and visual abstracts (i.e., either graphical or videotaped), which are by-products of computer-mediated communication. Bimodal (i.e., verbal and graphical) and multimodal abstracts (i.e., simultaneously verbal, graphical, voiced, or filmed with or without sound) are increasingly demanded by specialised high-impact journals and often displayed on YouTube or in journals' private video channels, separate from their hosting articles so as to be appreciated per se, as pieces with their own scientific and artistic value.

Graphical abstracts (henceforth GAs), which constitute the genre object of study in this article, are one of the two types of visual abstracts—the other one being video abstracts. GAs are also known as 'TOC (table of contents) images' and have been mistakenly thought to originate from Elsevier's 2010 'article of the future' project [2], while its true origin is to be related with the spontaneous fusion of three informative and promotional genres: the emblem, widely employed from the XV to the XVIII centuries, and the current infographic and advertising billboard [3].

The emblem, whose Greek name meant 'embossed ornament' or 'what is framed', consisted in an image (usually the portrait of a saint or some other prominent figure), framed by a motto above, frequently in Latin, and a textual commentary or brief legend at its foot. Its purpose was to provoke self-reflection on concepts, often allegorical, and biographies outstanding for their moral quality. Infographics display easy-to-understand images with some graphically dynamic components (e.g., vectors such as arrows or lines) that indicate the reading path (top-to-bottom, left-to-right, etc.) and confer the composition in a diagrammatic format which is very useful to explain processes or procedures. Last, billboard advertisements normally use a single static image taking up the whole promotional space with a minimum of verbal text (some slogan, the brand's name, or both) and sometimes none if the advertising brand is recognisable from the image alone.

Research on GAs has up to now addressed issues such as the dilution of genre moves and their effect on community boundaries [4], redundancy with the verbal text and rhetorical moves [5], media and discursive hybridity, emotion, and emergent scientist roles [6], authors and editors' perceptions [7], and metadiscourse and graphical design variants [3], all of which derive from the affordances of digital media.

Computer-mediated communication (hereafter CMC) has brought about a foregrounding of the scientist [8], emotional marketing and branding [9] and dissolution of boundaries between the private and public spheres, which has engendered a 'cyberspace college' [10] with a proximal communication code; that is, one based on the second-person and inclusive 'we' pronouns and deictics such as 'here' and 'now', which approach the sender, the scientific content and the communicative situation to the addressee. In this code, the traditionally sanitised academic discourse, which so many scholars have characterised as formal, depersonalised and factual [11–15], is acquiring a casual tone more fit for an informal conversation. Furthermore, the immediacy and resources offered by CMC have also raised the expectations of amusement in addressees and of promotion in institutions and corporations. The scientist's roles, in consequence, have come to include those of entertainer and advertiser, in addition to becoming a teacher and translator of technical concepts in the absence of science journalists. Moreover, as it is now required from scholars to transmit scientific content visually and multimodally, they are supposed to take on the role of graphic designers and filmmakers, unless they commission the task to professionals.

Despite these massive changes, Prior [16] (p. 520) notes that multimodality still remains a peripheral area of LSP (Language for Specific Purposes) research, as language has long determined topics and methods, and that the dominant questions posed by core journals in the field, such as *Journal of English for Academic Purposes* or *English for Specific Purposes Journal,* continue to be overwhelmingly language-focused. In this article, I set out to explore the challenges and problems caused by GAs and specifically by the phenomenon of visual stylisation, which may affect the implementation of more engaging dissemination models. In what follows, I shall describe the main features of the GA genre, provide a taxonomy of its visual stylisation and analyse its repercussions on the democratisation

of science. The focus on samples labelled as 'ineffective' by experts responds to the need to identify, with a view to their prevention, those graphical design practices that under expert eyes (specifically according to the criteria of science bloggers) obstruct the comprehension of scientific texts. A legitimate question that may arise is why not pay attention instead to those practices that 'work'. The answer is that the supposedly 'desirable' or 'effective' graphical options proposed by multinational scientific publishers, such as Elsevier, displayed on its website until 2020 and which I have classified at the end of the results and discussion section and illustrated in Figures 6–9, have not ensured correct interpretations among experts [4]. Complex reception studies should be conducted in each discipline with representative expert populations in order to ascertain the degree to which a given graphical design strategy is 'successful' in a particular context.

One may also wonder about the social role of science bloggers, as they appear to be the only members of the scientific community who publicly criticise the efficacy of GA designs across disciplines. They perform the function of 'whistle-blowers' warning other experts against the effects of particular graphical choices and showing them how these may generate miscommunication at two different yet intertwined levels: in the interpretation of the scientific message and the greater or lesser trivialisation of the interaction. Although science bloggers may contribute to the popularisation of specialised content and, along with it, to the democratisation of science dissemination, it cannot be assured that this is a consciously undertaken mission.

My contention is that visual stylisation cannot achieve more engaging and effective dissemination without attending to intercultural issues and complying with unified journal guidelines for authors, many of whom would need instruction in visual literacy and design. In other words, since interpretive skills depend in large measure on the addressees' cultural and scientific background and their acquaintance with the principles of visual representation, journal guidelines should inform of these principles, offer commented contrasts between desirable and ineffective practices, and thus stimulate reflexive creativity rather than achieving a unification of conventions through rigid graphical restrictions.

## 2. Methodology

The classification of stylisation types presented as outcome in this article results from the scrutiny of over 1000 GA samples from science blogs (slightly over 900 come from the archives of TOC-ROLF)[2] and research network forums, such as Academia.edu and ResearchGate, as well as from random findings in scientific journals (particularly from those specialised in the disciplines of Physical Chemistry and Chemical Physics) and GAs discovered by specialist teachers from my university (Universidad Politécnica de Madrid). To get an approximate idea of the current trends most criticised by science bloggers, I examined a total of 42 samples filed as arcane or trivial by the science blog TOC-ROLF between January and October 2021.

Although corpus representativeness is a construct, both theoretical [17] and methodological [18,19] and relies on intuition [20] since no 'pure corpora' can exist free from bias or theorisation, the samples in this study meet a threefold goal: first, they make the most recent GA corpus that could be compiled by the time this article was to be submitted. Second, they cover nearly a whole year of publication—the ten-month span from January to October 2021. Third, the 42 instances within that period suffice to give an idea of the various types of GAs criticised as ineffective in science blogs, and fourth, they come from several journals and disciplines. The predominance of GAs from Chemical Physics and Physical Chemistry over other fields is to be taken as naturally occurring data. Yet, it should be borne in mind that the elementary nature of their phenomena might pose less of a challenge for creative visual representation and metaphorical rendition than those from other disciplines, which may account for their abundance.

Multimodal analyses have been based on the principles and categorisations devised by Kress and van Leeuwen [21] and Machin [22], two of the most exhaustive categorisations within the panorama of Multimodality. The 42 samples possess graphical singularities that

belong to those categorisations and may justify their 'inefficacious' status, according to science bloggers. These multimodal parameters are strongly culture-bound, and their power of connotation is worthy of notice. The major classifying parameters are the following:

(1) Semiotics of colour

    a. Colour modulation (flat vs. nuanced);
    b. Tone;
    c. Hue;
    d. Saturation vs. dilution;
    e. Brightness;
    f. Luminosity;
    g. Differentiation (from monochrome to polychrome);
    h. Purity vs. Hybridity.

(2) Typography

    a. Weight (bolds vs. slimmer typefaces);
    b. Expansion (condensed or spread out characters);
    c. Slope;
    d. Curvature;
    e. Connectivity;
    f. Orientation;
    g. Spacing and alignment;
    h. Flourishes.

(3) Composition and panel layout

    a. Salience

        i. Cultural symbology;
        ii. Colour;
        iii. Tone;
        iv. Focus;
        v. Foregrounding;
        vi. Overlapping.

    b. Informative value through spatial placement

        i. Top/bottom positioning;
        ii. Triptych and centre/margin compositions;
        iii. Embedded structures.

    c. Framing

        i. Segregation;
        ii. Separation;
        iii. Integration;
        iv. Overlapping;
        v. Rhyming;
        vi. Contrast.

(4) Iconography

    a. Poses;
    b. Types of objects;
    c. Settings;
    d. Iconographic symbolism.

(5) Modality markers

    a. Degree in the articulation of detail

        i. Naturalistic vs. abstract;
        ii. Real-size or blown up;

    b. Degree in the articulation of backgrounds;

      c.     Interplay of light and shadow;

      d.     Visual depth.

(6)   Representation of social actors

      a.     Kinds of participants;

           i.     Individuals vs. groups;

           ii.    Anonymous;

           iii.   Archetypes.

      b.     Agency and action (roles)

           i.     Material;

           ii.    Behavioural;

           iii.   Verbal;

           iv.   Relational;

           v.     Existential;

           vi.   Mental.

      c.     Distance

      d.     Angle of interaction

           i.     Horizontal;

           ii.    Vertical;

           iii.   Oblique.

      e.     Gaze

           i.     Direct vs. indirect.

Metadiscursive parallelisms between the verbal and the visual have been drawn from Hyland's [23] taxonomy of interactive and interactional metadiscourse, which expands the notion beyond the mere textual scope into the interpersonal sphere. Colour, typography, composition and iconography may fulfill metadiscusive functions on either sphere and thus can transmit scientific content, while at the same time conveying emotion or connoting experience. On the interactive plane, it is interesting to discover what graphical devices are used as frame markers (in particular, as sequencers, stage labelers, topic shifters and transition markers or inferentials) to signpost the scientific narrative and discuss their efficacy. On the interactional plane, it is convenient to check if authors leave any personal imprint equivalent to a self-mention and whether the emphasizing (i.e., boosting) or mitigating (i.e., hedging) and attitudinal functions of colour, size, spatial collocation, typography, iconography (e.g., emojis) and expressive punctuation are intelligible and serve to enhance the message, and what engagement resources are exploited to engage readers/viewers. For example, question marks, lines, arrows, colour gradation and spatial collocation as directives; tropes, humour and cultural symbols as markers of shared knowledge or experience; and background or faded elements as potential asides or incidental disclosures.

Likewise, the concept of 'metaphorical scenario' has been borrowed from Musolff [24] to embrace metaphorical embodiments beyond the strictly notional into situational frames and behavioural patterns. Although the idea arose within the study of political discourse, its application is useful in other areas of human activity and natural phenomena. Visual tropes (mostly metaphors and metonymies) may reinforce cultural and disciplinary bonds through common cognitive schemata as markers of shared knowledge, perception and experience and regularly go hand in hand with humour. It is then necessary to see whether this association trivialises the scientific dissemination or helps to understand its content and grab and sustain the interest of viewers.

## 3. Findings and Their Discussion

### 3.1. First-Approach Perceptions: Major Challenges of the Graphical Abstract as Hybrid Genre

Regardless of their semiotic mode, host genre and format, abstracts have been defined as screening devices and previews of the research article [25], as selective representations rather than accurate summaries [26], and as promotional devices [26], even being metapho-

rised as the 'shop windows of science' [27] (p. 2). Visual abstracts could be labelled as 'hybrid' (see Bhatia [28,29]) or 'enculturated' research genres due to their intersemiosis and interdiscursivity. They merge creation and mimesis, the verbal and the visual, naturalistic and symbolic representations, and borrow elements from a variety of discourses, such as marketing and advertising, fiction literature, the graphic novel and the comic book, cartooning, photography and film. They are also multifunctional because they simultaneously encapsulate, engage, promote and serve as complex metadiscourse items. In this last respect, they may act as goal announcers providing the 'roadmap' of the article, as code glosses by means of embedded metaphorical narratives and analogies, as cognitive directives guiding the addressees' interpretation, as stage-labellers (if they consist in a single frozen image), and as attitudinal markers (depending on the type of rendition and artwork).

It is uncertain whether this enculturation or hybridity will result in acculturation, that is, in the total loss of idiosyncratic values and features due to the adoption of the dominant culture, in this case, that of marketing and advertising, or in a definitely failed democratisation of science dissemination. What seems clear is that, in visual abstracts, the deferential writer-responsible values of academic writing are giving way to a reader-responsible culture in which it is up to the readers/viewers to expose themselves to the stimulus or target information and focus their attention to finally recognise the message. This is the principle of the advertising–marketing culture [30–32], in which overt metadiscursive engagement markers (e.g., rhetorical questions, reader pronouns and mentions, asides, directives and expressions of shared knowledge in Hyland's 2005 taxonomy [23]) are not resorted to as much as in academia. Perhaps this displacement of responsibilities and orientations (from writer-responsible or reader-oriented to reader-responsible or writer-oriented) is the most remarkable discursive practice introduced by visual abstracts, which entail a threefold challenge (Figure 1): *transduction* [33], termed 'semiotic remediation' by Prior [16], *regenring* [34], called 're-purposing' by Prior [16], and *discourse economy*.

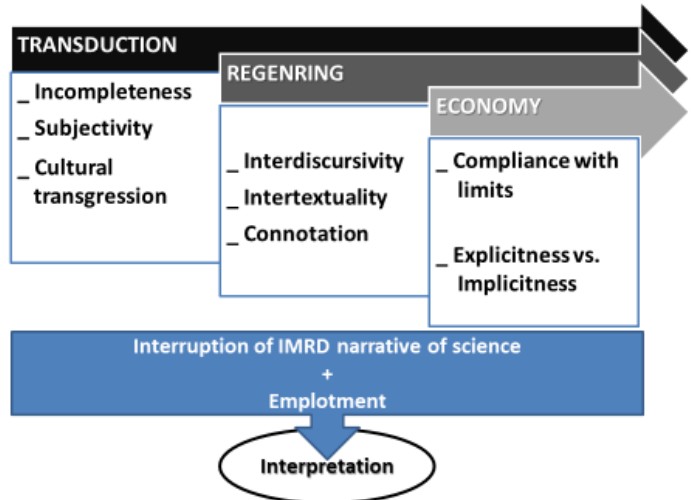

**Figure 1.** Threefold challenge due to visuality in GAs and video abstracts.

Transduction is the resemiotisation of content into a different mode of meaning: in the case of GAs, the conventional verbal summary of scientific articles, on average ranging from 100 to 300 words, is converted into an image or series of images fitting into a single panel. Regenring is a genre shift that modifies (either by clipping, by elaboration, or by some other type of manipulation) and reorganises that content in a way and with a purpose that substantially differ from the original aim, arrangement and/or extension. Discourse economy imposes extension limits, be they as a maximum footage duration for video abstracts, restricted space (frequently one panel) for GAs, number of points or aspects dealt with (so as not to clutter the space available for GAs and prevent cognitive overload in

video abstract viewers), file size, colour palette and saturation, and inclusion or exclusion of verbal text. All these limits and restrictions depend very much on journal policy.

The challenges inherent in transduction are *incompleteness, subjectivity* and *cultural transgression.* Static visuals are holistic and cannot express verbal content fully, so the information conveyed in a GA must necessarily be incomplete. Incompleteness normally impinges upon the representation of stages in the research process, as GAs showing a complete IMRD narrative sequence (i.e., Introduction–Method–Results–Discussion/Conclusion) do not abound. The addressee must then work out what moves are shown, which tend to be the methods and results stages in a sort of 'before' and 'after' narrative sequence. Yet the perception of qualitative evolution or of changes of state or condition is not always immediate because it may depend on minimal visual detail that requires full attention and not be guided by vectors (arrows or lines indicating the reading path) but by contiguous collocation. In such cases, as in sample (1), the cognitive processing effort is high and resembles the pastime activity of 'spot the differences' between two diagrams depicting two consecutive phases in the process of cellular resistance to anticancer agent doxorubicin. For the non-expert viewer or for experts with an untrained eye, such cognitive investment goes against the immediacy and efficacy expected from good summaries.

(1)  http://dx.doi.org/10.1016/j.bcp.2009.09.004 (accessed on 4 November 2021)
(2)  https://doi.org/10.1016/j.engstruct.2012.01.014 (accessed on 4 November 2021)

It may happen that the vectorless stage sequence consists of two disparate representations, as in sample (2), which contrasts a drawing sketch and a photograph of the same object of study (a beam joint) during a testing procedure. The legends explaining each image are not very telling, and neither are the 'frozen moments' captured in the visuals. These two samples are prototypical instances of compositional stylisation or layout subjectivity, which shall be further explored in a following section.

The selection of genre moves and steps, topical aspects and composition layouts is subjective and may go against the representational tradition of a given culture or even incur social taboo. A good case in point is Sample (3), which represents the properties of certain chemical elements as human behaviour. The resonance effects and affinity or bonding potential between zinc and selenium are metaphorised as a two-panel disco dance scene involving human silhouettes, where a female character in a mini-skirt (marked as zinc) dances provocatively to attract a male figure embodying selenium. This type of rendition may be not only misunderstood by certain cultures but also come across as offensive to particular religious creeds and gender collectivities with different mindsets and derived social roles, since the female character in (3) might appear as much too disinhibited or even lewd for some.

(3)  https://doi.org/10.1039/C5CP04498G (accessed on 4 November 2021)

In a similar vein, regenring may—intentionally or not—involve discoursal and textual loans and connote other interactions from distant spheres of human activity (e.g., journalism, art, fiction, advertising, comics, movies, etc.), but their perception is obviously conditioned by the addressee's cultural background and interpretive skills, particularly by metaphorical thought. Sample (4) is an instance of interdiscursivity with a discourse borrowed from outside science, concretely from Marketing and Advertising. The image in question verges on regenring because of its resemblance with a billboard or press advertisement, especially in its close-up take of the object/product to be promoted or paid attention to, and the involvement of the audience through a rhetorical question without any contextualisation and a direct appeal through the second person pronoun ('you'). In addition, the proximal tone achieved is characteristic with an 'exclusive we' pronoun as a marker of both nearness and authority.

(4)  https://pubs.rsc.org/en/content/articlelanding/2013/cc/c3cc44118k (accessed on 4 November 2021)

More conspicuously perceived than interdiscursivity, intertextuality aims at establishing 'universal analogies' by using visual and/or verbal references to widespread texts from literature, film, videogames, sports and other fields of human activity, as well as cultural icons and symbols and other presumably widespread life experiences. The choice of 'universal' references is per se subjective, despite the global reach of digital networks and other mass media: Can it be ensured that, say, an Asian or African scientist is fully familiarised with well-known works of art and literary and filmic canons from the western culture? And why should the western culture be the only referential basis for every connotation, analogy, metaphor and comparison used to disseminate and promote scientific achievements? Is it due to a population size criterion or to a matter of power?

It cannot be denied that, by and large, science and technology are practically in the hands of a few highly industrialised countries with a Judeo–Christian origin, but that should not oblige experts with other backgrounds to acquire the cultural level necessary to decode scientific content in a genre whose major purpose is screening and encapsulating, not entertaining. However, messages praising the aesthetic and amusing qualities of visual abstracts, such as that launched in 2018 by biologist and journalist Kerry Evans (Senior Managing Editor for *AJHG* and *Immunity* and in charge of the Cell Press blog 'Crosstalk'), have fuelled the idea that entertaining is part of their objective. She holds that there is no reason why Cell's video abstracts cannot be enjoyed by lay readers/viewers:

> "Just because Cell video abstracts are primarily intended for scientists doesn't mean your kindergartener or grandparent can't enjoy this." [35]

To the discourse analyst, nonetheless, entertainment without content comprehension is difficult to imagine, and chances that scholars gloss their scientific texts for laypeople are really thin unless the intention is to produce an outreach version. The publishers' view of scientific communication between experts as an amusing interaction seems then to clash with that of scholars, for whom it is a concise, informative transaction with a high degree of taken-for-granted knowledge.

While the verbal abstract is infallibly literal, monosemic and discourse-specific, its graphical and video variants are allowed to be metaphorical and interdiscursive, even polysemic for the ideal audience, depending on the degree of literalness adopted. This fact might lead us to think that the intended addressees of verbal and graphical abstracts could be different, although no scholar has as yet disclosed an intention to reach out to laypeople by means of GAs. Sample (5) expresses scientific doubt with the Shakespearian plot of *The Tragedy of Hamlet, Prince of Denmark.* As shown in (5), the 'ad hoc pointer' [36] or 'metaphorical trigger' that signals the metaphorical nature of the encoding is the molecule held by the character dressed in a 16th-century attire, who should instead hold a skull, according to the Shakespearian play. The verbal and phonic pun of the title "tBu or not tBu?" (a question which emulates Hamlet's famous soliloquy "To be or not to be, that is the question") is another pointer, tBu being the name of the ligand molecule studied. In the verbal paraphrase of the GA facilitated by the journal (very few publishers do), the authors describe their article as 'a Hamlet study', a qualification that contrasts starkly with the technical register used in the remainder of the paraphrasing paragraph.

(5)   https://doi.org/10.1002/chem.201102674 (accessed on 4 November 2021)

Intertextual choices likely to be less 'universal' are (6), an allusion to the sentence "Quo vadis?" from the Acts of Peter, one of the earliest apocryphal Acts of the Apostles in Christianity and later on inspiration for Henrik Sienkiewicz's 1896 novel *Quo Vadis* , and the national folklore instances showcased by (7) and (8), which respectively embody Aesop's fable *The Hare and the Tortoise* and the popular idiom 'carrot-and-stick', used to denote a dual motivational approach consisting in reward and punishment. In both samples, the authors have spared any verbal clue, which indicates that they regard their cultural referents as accessible enough.

(6)   https://doi.org/10.1007/10_2018_75 (accessed on 4 November 2021)
(7)   https://doi.org/10.1039/C5PY01964H (accessed on 4 November 2021)

A yet more subtle level of decodification based on evoked concepts or situations/scenes is connotation. Sample (8) parodies the TV programme *The Joy of Painting*, broadcast between 1983 and 1994 in the USA by PBS. In it, artist Bob Ross (1942–1995), soon a celebrity, painted canvases live in less than 30 min to explain diverse rapid pictorial techniques. The palette held by his caricature in this GA contains only cobalt blue dollops, and his paintbrush represents the zinc element, with which the brushstrokes on the canvas seem to be applied with extreme smoothness. The artist's characteristic permed hairstyle is here hexagonally shaped, iconic of the catalyst molecule under research. The processing of the information conveyed in this GA is not easy for scientists unexposed to the said TV programme, whose metaphorical scenario is not directly related with the research topic, but laterally suggested by the shared features of rapidity and simplicity: the cobalt–zinc catalyst is as quick and smooth as Ross' painting. How many researchers from outside the USA are able to recognise the caricature and then deduce this implicit association? Undoubtedly, Bob Ross has had an impact beyond the U.S. frontiers because, curiously enough, the GA authors are four German scholars based in Munich and a Chinese national affiliated to Wuhan University, but surely Ross' popularity will vary across countries and his figure may be unknown in some of them around the globe. It remains uncertain how the five authors became acquainted with the celebrity and whether they took for granted a 'universal' audience for their GA. Can this case be considered an instance of cultural colonisation or imposition within the scientific community?

(8)    https://doi.org/10.1002/cctc.201901939 (accessed on 4 November 2021)

To conclude, the compliance with extension limits and the primary goal of summarising content brings along a discursive economy in the form of omission (of information taken for granted or considered superfluous) or implicitness (information subtly hinted at in a visual manner). Variations on the hare-and-tortoise motif containing only one of the two characters in the story, most often the victorious turtle (see Sample 9), exemplify the strategy of omission and assume that addressees know the plot, which makes it unnecessary to include the defeated hare. Sample (10) is another mixture of omission and implicitness: the addressee's knowledge of the chemical arrangement of the nanoparticles under study theoretically suffices for a successful interpretation of the image, although it may not be so in practice.

(9)    https://doi.org/10.1002/chem.201103973 (accessed on 4 November 2021)
(10)   https://pubs.acs.org/doi/abs/10.1021/jp407495z (accessed on 4 November 2021)

So far we have seen how the three main challenges generated by visuality in science dissemination (i.e., transduction, regenring and discursive economy) frequently cause the interruption of the IMRD narrative sequence of GAs or intervene in it and lend themselves to embedded emplotments devised to epitomise, clarify through metaphorical analogies and seek memorability of the scientific content, all of which affect interpretation. We have also had the opportunity to observe that in many instances there is a stylised or subjective presentation of the visual message. Let us now turn to the definition of the stylisation phenomenon and its repercussions upon science dissemination.

*3.2. Finer-Grained Results: A Working Definition of Stylisation, Taxonomy and Outstanding Issues*

Stylisation is not all about embellishment, although part of it may be aesthetically motivated. I define the concept as a subjective encoding of information, in this case scientific, out of pragmatic and aesthetic reasons, which frequently leaves an authorial imprint and affects the comprehension of the message. Stylisation has to do with individuals' creativity and subjectivity, since it does not strictly follow external guidelines and norms, may traverse all the three major challenges posed by visuality (i.e., transduction, regenring and discourse economy—see again Figure 1), and even operate metaphorically by recounting scientific and technical facts with embedded non-scientific narratives. In these, recourse to intertextuality, interdiscursivity metaphorical scenarios and connotation (the latter in

more subtle encodings) is common. Some instances have already been shown in samples (3)–(10).

Hence, stylisation may deal with notional, interactional and compositional aspects alike (i.e., roughly corresponding with the functions of language put forth by Systemic Functional Linguistics: ideational, interactional and textual) and exhibit varying degrees of complexity that range from mere ornamentation to multiple metaphorical embodiments, sometimes nested. Figure 2 shows a taxonomy proposal that divides stylisation into 'simple' (essentially consisting in embellishment) and 'complex' (conceptualising facts or phenomena).

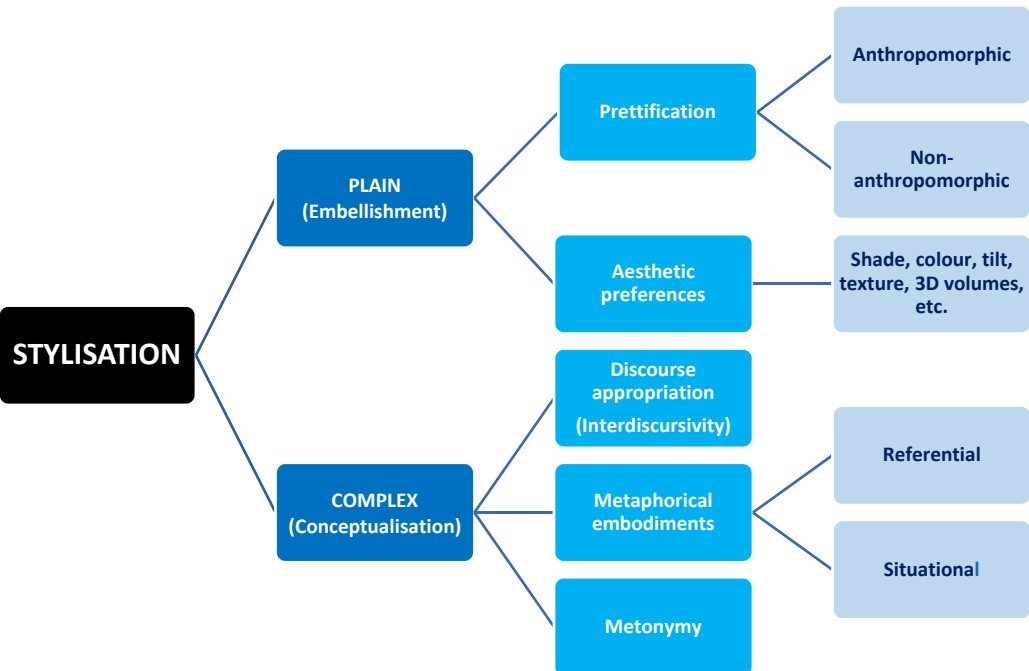

**Figure 2.** A proposed taxonomy of stylisation in GAs.

Within plain stylisation (i.e., embellishment), 'prettification' is the term used by science bloggers to denote the subjective representation of chemical elements and compounds, atoms and molecules, microorganisms, animals, plants and processes as animate beings, among which anthropomorphic shapes and facial expressions are rife. Samples (11) and (12) are instances of anthropomorphical prettification. While the one in (11) does not seem to affect the interpretation of the scientific information, that of (12) might contribute to a more accurate comprehension since the prettified molecule has additionally been metaphorised into a human archer aiming his/her arrow at a chemical ($O_2$) target, which suggests agency and accuracy on the part of the substance prettified. A case of non-anthropomorphic prettification is (13), where the sweetener molecule synthesised is metonymically rendered as ice cream on a wafer cone.

(11) https://doi.org/10.1021/co500146u (accessed on 4 November 2021)
(12) https://doi.org/10.1039/C3CC47261B (accessed on 4 November 2021)
(13) https://pubs.acs.org/doi/abs/10.1021/jf301600m?mi=14f7i14&af=R (Accessed 30 November 2021)

It is not common to find visual metonymy in isolation. Very occasionally, it is used to represent actions that integrate the methods stage in the research and epitomises those actions with the instruments or tools with which they are performed: for example, a syringe for inoculation, a stove for drying, a tap for water rinsing, or a clock for waiting time. Metonymic 'purity', anyhow, is a debatable concept, as all the former instances

could be conceived as metaphors under the conceptual schema AN ACTION IS ITS INSTRUMENT/TOOL.

Prettification, oftentimes aided by cartooning and comic book techniques, brings about register shifts that trivialise the communication or turn it into an informal interaction. In (14), two molecules have human-like bodies and, although faceless, speak and their utterances appear contained in speech bubbles, whereas in (15), a thought balloon is attributed to an insect. Certainly, these kinds of stylisations do not hamper the democratisation of science: they do not segregate readers/viewers nor hinder the grasp of scientific content but contribute to ingraining the goal of entertainment in science dissemination, which might end up threatening interpretation if pursued at all costs.

(14) https://pubs.acs.org/doi/10.1021/jp4081977 (accessed on 4 November 2021)
(15) https://www.mdpi.com/2072-6643/5/5/1622 (accessed on 4 November 2021)

Aesthetic preferences involving colour, shading, image orientation, texture, or three-dimensional effects are potentially less innocuous than prettification, as Sample (16) demonstrates. The arbitrary shading of its molecule segments may lead to a mistaken interpretation of the shaded areas as bonding surfaces, an ambiguity criticised by science bloggers.

(16) https://onlinelibrary.wiley.com/doi/abs/10.1002/anie.200904588 (accessed on 4 November 2021)

All in all, then, the chief dangers of stylisation, be it strictly graphical or conceptual, are trivialisation, exclusion and misinterpretation—the latter two repercussions even among experts. Displayed in Figure 3 are the resources causing the GAs selected from the TOC-ROLF science blog to be deemed ineffective in their summarising, screening and disseminating function.

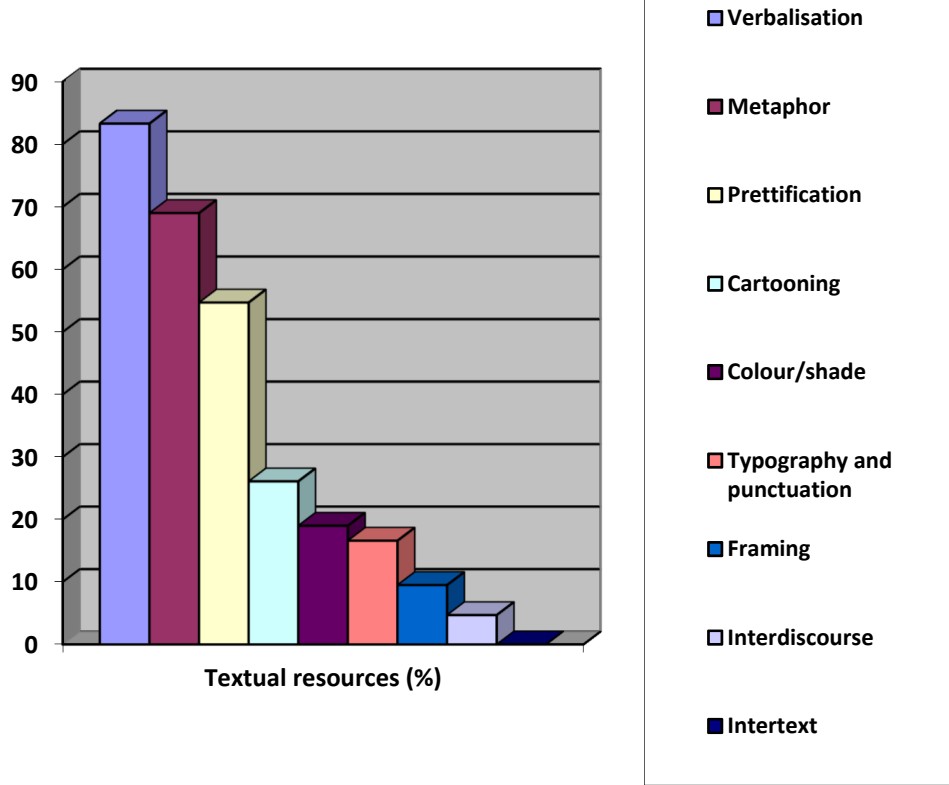

**Figure 3.** Percentages of the textual resources used in the TOC-ROLF blog's 2021 samples (January–October).

Noticeably, there is a predominant use of verbal insertions (83.3%), mostly of succinct noun phrase labels (73.8%). By contrast, verbalisations comprising one or more sentences are relatively scarce (9.5%) and associated with speech bubbles and thought balloons,

two cartooning techniques that inevitably shift the academic register into an informal interaction. Other techniques present are creative panelling, spiky balloons to denote loud sound, clash or violent reaction, explanatory captions and movement runes. Slightly over half of the samples (54.7%) prettify human characters, animals, objects, molecules and chemical elements or substances, and somewhat less than 20% make use of shading and colour saturation, which also alters the register. Only two instances appropriate the visual discourse of Advertising: one with a close-up photograph of a glass of beer with the printed question "How many bubbles?" as the hook, which clips the engaging title of the article "How many $CO_2$ bubbles in a glass of beer?" (17):

(17)  https://doi.org/10.1021/acsomega.1c00256 (accessed on 4 November 2021)

and the other (18) with a full-blown mimic of a cow that intriguingly resembles that of the 'La vache qui rit' brand icon[3], of the Groupe Bel food firm:

(18)  https://doi.org/10.1021/acs.inorgchem.9b03563 (accessed on 4 November 2021)

Finally, also worthy of attention are the facts that recourse to intertextuality is non-existent and that metaphor is the second most used device (a little below 70%), but with a prevalence of object- or concept-based mappings (45.2%) over scenario-based ones (23.8%) in the source domain. None of the science metaphors detected evokes works of art, literature or film but prosaic life experiences applied to the target entities or processes to be described, or extremely unrealistic situations created ad hoc. Some examples of each variant are presented in Table 1, which rates their proximity or familiarity to the addressee ('commonplace' versus 'unrealistic' objects and experiences). Regardless of these two degrees of proximity, however, there is always a 'metaphorical trigger' or 'ad hoc pointer' [36], an incongruous element that beacons the need for not interpreting the embodiment literally. Prettification itself, some bizarre visual associations (e.g., a string coming out of a molecule for a cat to play with, perhaps representing bonding lengths), the insertion of scientific matter (e.g., chemical formulas and mathematical calculations, molecular models and microscopic images), verbal labels, or even colour saturation and whimsical framings are but a few of the devices serving as triggers.

In the context of science and technology dissemination, necessarily monosemic, economic and linear in the rhetorical organisation of its specialised discourse, many interpretive difficulties at a structural level reside in the motivation of the metaphorical embodiments of abridged visual texts, such as GAs and video-abstracts. It is relatively easy, therefore, to identify metaphorical triggers and the elements that are 'out of place', but sometimes it is rather complicated to find out the relationship between the source and target entities, no matter the amount of technical expertise. Occasionally, the covert motivation is subtly metonymic, as in the GA (mentioned in Table 1) showing a prettified Santa Claus-like molecule of holmium oxyhydride at a sunny beach to suggest thermal stability. The relationship has to do with the stoichiometric composition (HoHO) of the substance, reminiscent of Santa's typical laughter (19):

(19)  https://doi.org/10.1021/acs.inorgchem.0c03822 (accessed on 4 November 2021)

In another of the samples (20), also referred to in the table, an octopus-like chemical compound with a hat on to protect its ionic head, is sitting on a gold ingot, in front of several switches. The whole scene refers to the compound's multiple ligand properties, equated with eight tentacles, particularly with gold and with a view to developing nanoscale molecular switching materials:

(20)  https://tocrofl.tumblr.com/post/666579375759753216/octopus-in-hat-switching-switches (accessed on 4 November 2021)

**Table 1.** Some salient examples of object-based and scenario-based metaphorical embodiments in the blog samples studied (TOC-ROLF January–October 2021).

| Trigger | Object | Proximity | Scenario | Trigger |
|---------|--------|-----------|----------|---------|
| Chemical formulae above and beside exclamation mark | Multilayered sandwich (*Suggesting compactness and internal structural features*) | COMMONPLACE | Fishing | Fish-shaped molecules |
| Chemical formula on it | Glazed doughnut (*Relevance unknown without reading full article*) | | Cat playing with string | String attached to solid molecular model, animal prettification |
| Prettified molecule inside | Cage (*Embodying ionic sequestration*) | | Job hunting | Prettified molecule in suit and with briefcase, holding its CV |
| **Source-domain mappings in object-based metaphorical schemata** | | UNREALISTIC | Santa Claus at the beach | Prettified molecule, colour saturation |
| | | | Meal guests along an industrial conveyor belt | Prettified molecules, weird etiquette |
| | | | Octopus with hat on and seated on gold ingot in front of switches | Prettification, formulas and graph around |

This small-scale analysis brings to light the complex metadiscursive multifunctionality of GAs: at a macrolevel and according to Hyland's categorization [23], they work as engagement markers (i.e., as attention-getters and cognitive directives guiding the addressees' interpretation) and as glosses (via some vectors, didactic metaphors and embedded narratives parallel to or interrupting the rhetorical IMRD pattern of science dissemination). They also function as evidentials, if they consist of visuals displaying results from the original study and serve as goal announcers when inserted in the research article between the authors' names and affiliations and the introduction. At a microlevel, they may contain a wide array of visual metadiscourse items in tension between mimesis and expression, a rooted feature of science iconography [37]. Many such items coincide with the elements enumerated by Kress and van Leeuwen's attempt [21] at building a visual grammar and with Machin's prompts for multimodal analysis [22], both of which comprise vectors (arrows, lines, runes, etc.), frames, collocation, sizes, light and chromatic effects, typography, perspective and angles of interaction, and degrees in the articulation of detail. This coincidence of resources, most probably, will be the result of intuitive choices on the scholars' end, who very seldom receive any institutional training in visual design.

In light of all the former, it is reasonable to expect that repertoires vary across disciplines and in the end position scholars and research areas with regard to the scientific message and its intended audience. A skimming look through the TOC-ROLF archive since its earliest entries in August 2010 reveals that some motifs are 'endemic' to certain scientific fields, such as the use of Aesop's hare and tortoise fable to qualify the efficacy of catalysers and chemical reactions in Physical Chemistry and Chemical Physics. Storylines of this kind (provided by metaphorical scenarios from literature, film and folklore) constitute, together with speech acts (especially interrogative and commissive ones, in the form of rhetorical questions and verbal or visual commands) and past and imposed positions, the three pillars of any positioning action, which are known as 'positioning cluster/triangle' [38]. By 'past positions' we should understand the scientists' dissemination trajectory in publishing

and public speaking, with all the visual resources they have employed thus far out of personal preference in GAs, in-article illustrations, and slides for lectures and conference presentations. By 'imposed positions' we should interpret the vetoes, restrictions and encouraged options (e.g., exemplars, templates, imitation trends in scholarly circles and author guidelines in general) fostered by the discipline or field and by scientific journals and conference committees.

Logically, the recording of individual positioning turns very difficult, because it implies tracking the 'visual biography' of each researcher diachronically along his/her academic career and across different genres, whereas the synchronic description of disciplinary and editorial positions, of collective 'visual identities', is more feasible. On the other hand, and as happens with verbal metadiscourse items in Hyland's taxonomy [23], certain visual resources may be overlappingly perceived as members of different categories. Arrow-shaped vectors, for instance, may be decoded as interactive and interactional at the same time; that is to say, as sequencers and cognitive directives (engaging commissive markers), whereas the expression of dissuasion, disapproval or rejection of actions not to be performed in a particular scientific procedure are more explicitly and unambiguously marked by sad or angry emoji gestures, red crosses and prohibition signs. Just like verbal metadiscourse, in sum, one item may fulfil several functions and one function may be performed by several forms, which in GAs jeopardises interpretation even more than in the verbal encodings of conventional abstracts, given that visuals are by nature holistic and schematic and GAs offer very little room for verbal inserts and proper contextualisations.

The disciplinary provenance of the TOC-ROLF samples analysed is quite homogeneous. Out of the 22 journals represented in the ten-month corpus, the majority was found to address the global chemistry community or to belong to broad subdisciplines, such as Inorganic and Organic Chemistry (Figure 4). The established combined disciplines of Chemical Physics and Physical Chemistry (9% of corpus instances) and multidisciplinary journals (14%) covering a wide-ranging spectrum of (sub)disciplines at the interface of Materials Science, Chemistry, Engineering, Physics, Medicine and Biology yield not very disparate smaller proportions, as can be seen in the pie chart of Figure 4. The titles of the journals in question are listed and sorted out by discipline in Table 2, which shows an overwhelming presence of the ACS Publishing Center, followed in equal proportion by Elsevier and Wiley, and at a small distance by the Royal Society of Chemistry, based in the U.K. Springer Publishing, in contrast, trails behind with only one sample. A focus on the percentual weight of each publisher's presence in the corpus is shown in Figure 5.

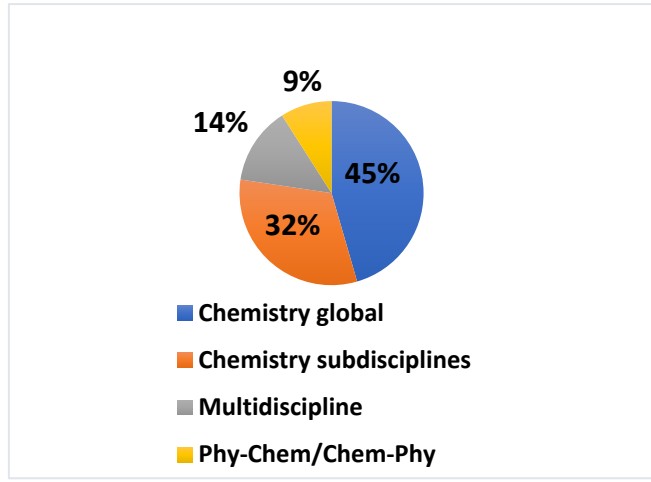

**Figure 4.** Percentages of samples' disciplinary provenance.

**Table 2.** Journal titles and publishers classified by discipline.

| Discipline | Title | Publisher |
|---|---|---|
| GLOBAL CHEMISTRY | *Accounts of Chemical Research* | ACS |
| | *ACS Omega* | ACS |
| | *Chemical Reviews* | ACS |
| | *Chemical Science* | RSC |
| | *Chemistry. A European Journal* | Wiley |
| | *Journal of the American Chemical Society* | ACS |
| | *Journal of Chemical Theory and Computation* | ACS |
| | *Monatshefte für Chemie-Chemical Monthly* | Springer |
| | *Small: Nano-Micro* | Wiley |
| | *RSC Advances* | RSC |
| SUBDISCIPLINES | *ChemCatChem—The European Society Journal for Catalysis* | Wiley |
| | *EurJOC—European Journal of Organic Chemistry* | Wiley |
| | *Green Chemistry* | RSC |
| | *Inorganic Chemistry* | ACS |
| | *Journal of Organic Chemistry* | ACS |
| | *Journal of Organometallic Chemistry* | Elsevier |
| | *Tetrahedron* | Elsevier |
| CHEMICAL PHYSICS and PHYSICAL CHEMISTRY | *The Journal of Physical Chemistry* | ACS |
| | *The Journal of Physical Chemistry Letters* | ACS |
| MULTIDISCIPLINARY | *ACS Macroletters* | ACS |
| | *Environmental Pollution* | Elsevier |
| | *Journal of Functional Foods* | Elsevier |

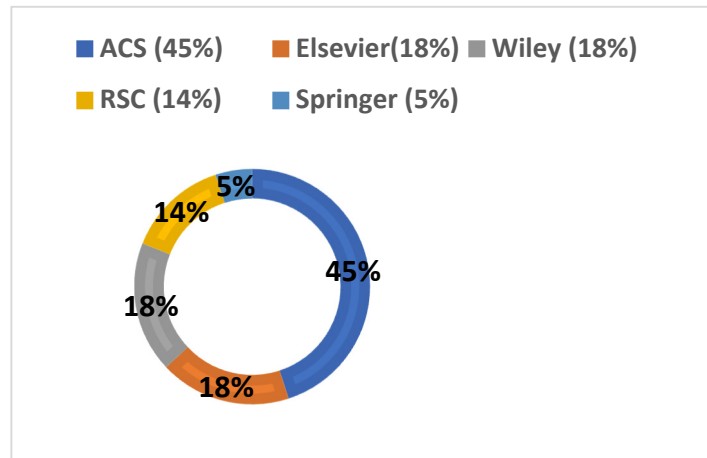

**Figure 5.** Percentages of publishers' presence in the corpus.

Although the larger presence of a particular publisher (see Table 2 and Figure 5) might suggest vaguer instructions to authors, it is not necessarily so, as other factors may come into play, such as journal impact, reviewing times, percentages of rejection, or topic currency and centrality. Springer's *Monatshefte für Chemie-Chemical Monthly,* for example, is represented by one single sample and does not issue specific guidelines for GAs, just

a 'template for chemical drawings' tackling aspects such as 'line art', 'colour art', 'figure captions', 'figure lettering and numbering', or 'figure placement and size'.

The rest of publishers' policies are more concrete and referred to GAs, also called 'ToC image', 'ToC graphic/figure', 'ToC entry', or 'graphical ToC' (ToC being the abbreviation for 'table of contents'), yet specificity may increase in the guidelines for authors of each of their journals. ACS Publishing Center, nevertheless, just issues a set of general instructions accessible through a link in each of its journals and remarks that the graphic should be simple, informative and understood by someone who has not read the manuscript. It may consist in a structure, a graph, drawing, photo, scheme or a combination, and should avoid long phrases or sentences, as well as the inclusion of photographs and caricatures of any person, living or deceased. The use of colour is encouraged (an instruction shared by all publishers), and the standards of scholarly professional publications must be met, although they are not spelled out.

In its general guidelines, RSC includes a series of 'do's' and 'don'ts' that limit verbal text to 15–25 words and graphical elements to a maximum of two, recommending focusing only on key findings and using easily recognisable words that can be read quickly. Among the don'ts are the repetition of information present in the title and the use of graphs, spectra and too much detail. Its journal *RSC Advances* adds the specifications that the visual must comply with the principles of political correctness and not contain logos, trademarks or brand names, the text supplied must be one or two sentences long and comprise a maximum of 250 characters, and the graphic should 'capture the reader's attention' (without any provision of compositional resources).

Wiley is the only publishing company that presents potential authors with the initiative of commissioning the GA to a professional in-house designer, and provides three exemplars respectively structured into one, two and three panels to show possible outcomes. Its general guidelines for authors detail the graphic's dimensions, the type of file, copyright issues, and vaguely encourages selecting a figure that 'best represents the scope of the paper'. More restrictive instructions are those issued by its journals: when dealing with political correctness, *ChemCatChem* and *EurJOC* allow recourse to elements of mythology, legends and folklore, which might be accepted on a case-by-case basis, but their policies discourage the use of religious iconography and imagery and of any object with a cultural significance. *Chemistry. A European Journal* discards, in addition, needless shading and asks authors to check the journal's policy of colour use. The fourth of the Wiley journals, *Small,* demands that the accompanying verbal text does not exceed 60 words, is written in the third person and for a 'general audience', a rather fuzzy requisite within such a stringent condition.

The four Elsevier samples in the corpus will surely have followed the guidelines for GAs in the publisher's website until 2020. According to them, the visual should allow readers to 'quickly gain an understanding of the main take-home message of the paper', 'encourage browsing' and 'promote interdisciplinary scholarship'. It is assumed to represent 'the work described in the paper' and adopt any of the compositional patterns found among the 16 exemplars shown online during the time span 2015–2020, whose basic typology is summarised in Figures 6–9.

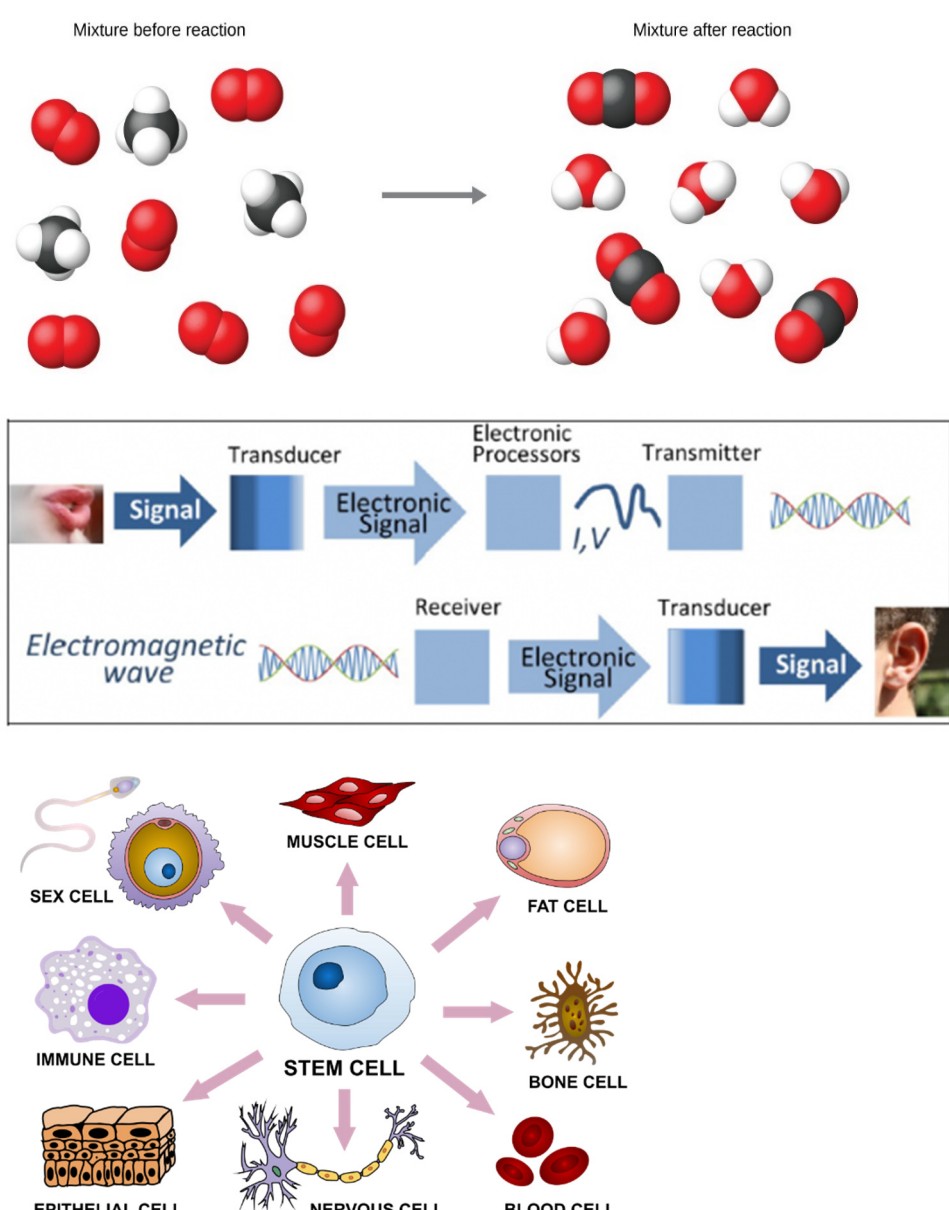

**Figure 6.** Arrows as narrative (**top**), reading path (**centre**) and classificatory (**bottom**) devices [39–41].

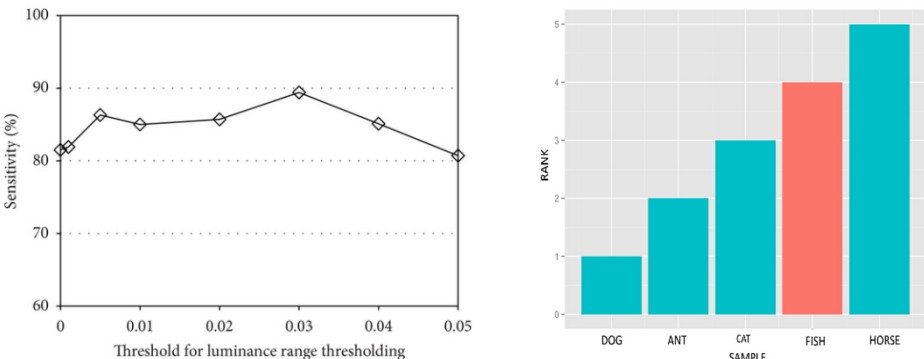

**Figure 7.** Some instances of 'data displays' [42,43].

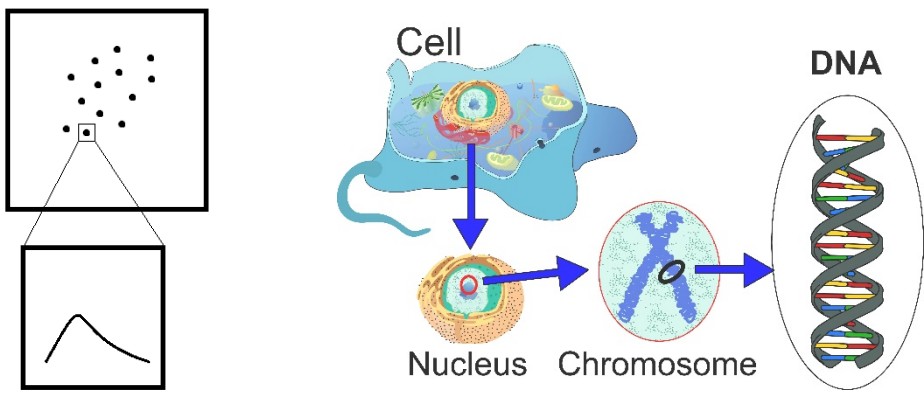

**Figure 8.** Zoom-ins for detail foregrounding [44,45].

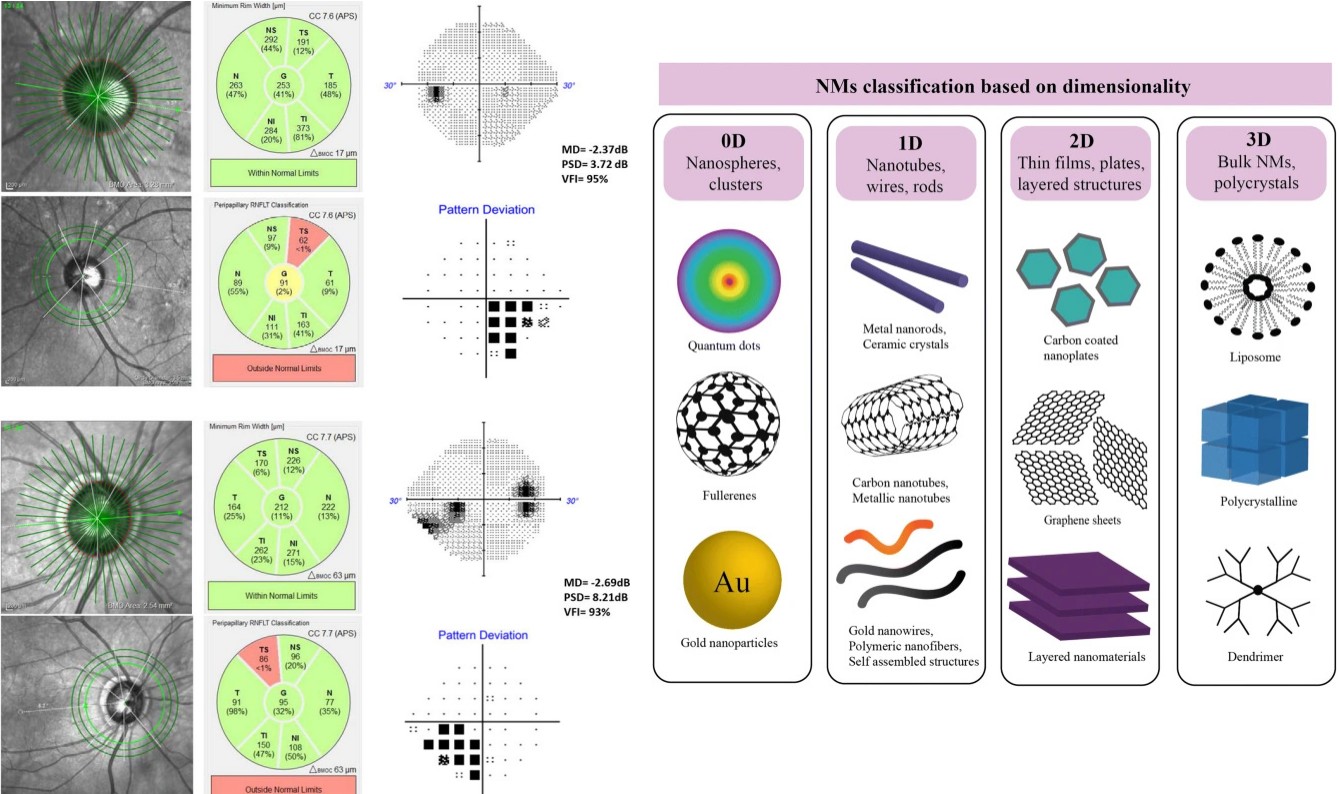

**Figure 9.** Juxtaposed collocational arrangements for classification, narrative and zoom-in combination [46,47].

Figure 6 gathers three vectorial compositions based on the use of arrows (replaceable by lines). Vectors (here arrows) can be seen to perform three major functions: mark a narrative in which there has been some change in state or condition (i.e., differentiate a 'before' from an 'after'), signal the reading or viewing path through the several stages of a process or procedure, and classify items from a superordinate category, indicating their derivation or exemplifying function.

Visual displays of various sorts (Figure 7) make up another frequent compositional pattern that entirely depends on editorial policies. Visual data in the form of graphs/charts and diagrams may appear as a minor element within a larger pattern or constitute a major one themselves, although such cases do not abound owing to the ever-increasing request for originality and creativity on the part of publishers and journal editors. This requirement tends to exclude the reproduction of graphics present in the article and to foster

eye-catching compositions, but few journals caution against the risks of certain practices adopted for that purpose, such as colour saturation and cluttered visual arrangements. Elsevier's *Cell* is one of those few journals.

With or without vectors, the amplifications of detail, which I termed 'zoom-ins' in a previous work [3], normally lean on collocational and framing resources and are reserved for the display of substructures or phenomena invisible to the naked eye. Both illustrations in Figure 8 employ stylised lense icons as frames, but the mere collocation of an unframed finer-grained picture usually suffices to understand the visual progression.

Last but not least, classificatory arrangements through juxtaposed collocations and with or without verbal labels (Figure 9) are an alternative to tree diagrams, although they may be mistaken for vectorless collocational narratives. The category or type, usually on top position, may be represented by a verbal label or a visual (labelled or not), below which subcategories and examples are placed. The same organisation may apply for progressive zoom-ins and narrative denouements in successive phases.

Since the beginning of 2021, Elsevier has changed its GA submission policy and demanded a unique compositional scheme consisting of three consecutive panels (see Figure 10) for the rhetorical moves of 'research contextualisation', 'methodology' and 'outcome', in order to curb the insertion of extraneous material and subjective metaphorical conceptualisations and set more universal graphic conventions.

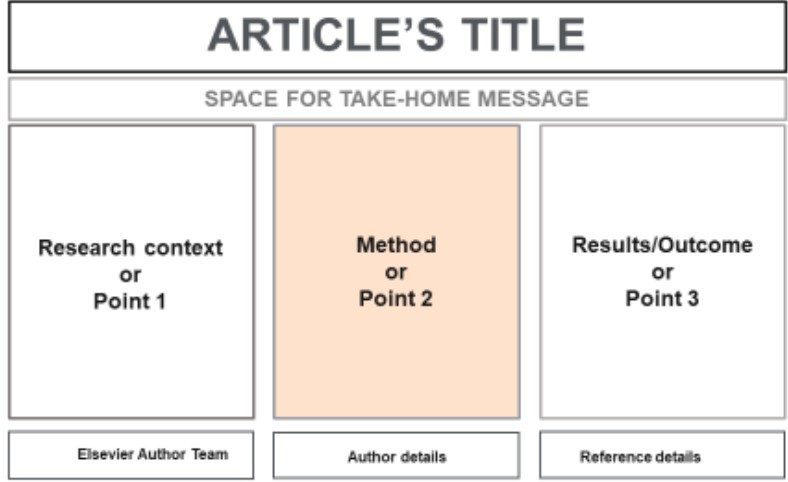

**Figure 10.** Elsevier's three-panel compositional proposal (rendition of the publisher's free-download original).

The idea is that readers/viewers grasp the article content 'at a single glance', but the four exemplars provided are not as lean as they should be for immediate visual capture. Only time will tell whether this goal will be eventually attained and whether modality markers and metaphorical embodiments will permeate the tripartite frame.

## 4. By Way of Conclusion: Towards Established Rhetorical Stylisations?

Elsevier's recent option for a three-panel model as single compositional pattern seems intended to fight stylisation through limitation and simplification, in the pursuit of more objective and universal patterns. The compositional arrangement chosen functions as an interactive metadiscourse item that marks three distinct stages in the scientific IMRD narrative and indicates the reading path to follow (left-to-right) without the aid of vectors. Yet this fair and economically democratising measure has its flipside: if compositional layouts express authorial voice, then the imposition of a single rhetorical scheme by publishers and journal editors may be accused of stifling individual creativity and dissuading authors from exploring pictorial possibilities that might generate knowledge. We must not forget that, in

Kress' words, "representation makers are knowledge makers" [33] (p. 27), and learning is "a dynamic process of sign making" not necessarily dependent on verbal language.

Tripartite exemplars, moreover, remain uncommented and, in consequence, scholars cannot learn from their design weaknesses nor incorporate valuable strategies from others to their own graphical repertoires, at least at a rhetorical macrolevel. Granted, templates may become 'editorial rhetorical stylisations' that define a journal or a publisher's identity, but at the expense of authorial choices and perhaps of the discipline's visual distinctiveness, because not all journals reflect what scientists do when disseminating knowledge and journal guidelines are not always followed by scientists. Furthermore, the very moment a stylisation becomes established and used across the board, it ceases to be a stylisation and evolves into a standard, from which new stylisations may stem.

A threefold challenge lies then ahead for further research: first, it will have to find out if the visual rhetoric conveyed by this tripartite template curtails subjectivity in favour of textual comprehension, and thus helps democratise scientific content. A full understanding of this content by lay audiences is, of course, asking too much in many disciplines, but it is not impossible to increase clarity for experts from akin fields and for communication and discourse analysts. Second, it will be necessary to investigate how much influence the template exerts among related journals, which might adopt the same model and turn it into a 'disciplinary rhetorical stylisation' (rather standard) and tool. The imposition, circulation and consumption of discourses always involve power issues, in this case, intra- and interdisciplinary, between scholars and their audiences (intended and untargeted), and between multinational publishing giants and the scientists' population. Third, visual corpora will need to be scrutinised to determine whether the newly acquired template will keep free of distracting elements and subjective modality devices that go against clarity, economy and immediacy. Let us wait and see.

**Funding:** This research received no external funding.

**Data Availability Statement:** Data are contained within the article.

**Conflicts of Interest:** The author declares no conflict of interest in the publication of this article.

## Notes

1　https://www.peta.org/blog/umass-simulates-hot-flashes-in-marmosets/ (accessed on 4 November 2021)
2　https://tocrofl.tumblr.com/ The acronyms stand for 'Table of Contents-Rolling on Laughing Floor'. This blog has been inspired in turn by other blogs in the science news outreach magazine *Discover* (https://www.discovermagazine.com/; http://blogs.discovermagazine.com/discoblog/category/ncbi-rofl/) (accessed on 4 November 2021)
3　https://tocrofl.tumblr.com/ The acronyms stand for 'Table of Contents-Rolling on Laughing Floor'. This blog has been inspired in turn by other blogs in the science news outreach magazine *Discover* (https://www.discovermagazine.com/; http://blogs.discovermagazine.com/discoblog/category/ncbi-rofl/) (accessed on 4 November 2021)

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
