# Peer review of "Scientific Stylisation or the ‘Democracy Dilemma’ of Graphical Abstracts"

_publications, doi:10.3390/publications10010011_

Round 1

Author Response

Thank you for your comments. We revised the paper. Please see the attachment.

Reviewer 2 Report

General impression
In the context of information overload, this topic seems to be very relevant and interesting; so the reader(s) will find a short visual summary of the content.

Introduction
In the introduction, it would be a good idea to give a broader view on graphical abstracts and to cite the following reference:

JungWon Yoon, EunKyung Chung (2017). An investigation on Graphical Abstracts use in scholarly articles. International Journal of Information Management, Vol. 37, Issue 1, Part A. p. 1371-1379. https://doi.org/10.1016/j.ijinfomgt.2016.09.005.

Please give the meaning of the term «LSP» in line 69 (languages for special purposes?)

Methodology
«random findings in scientific journals (particularly 85 from those specialised in the disciplines of Physical Chemistry and Chemical Physics)", lines 85, 86: Why is there a limitation to these disciplines?
Please explain this.

Terminology:
Please make a clear distinction between «visual» and «graphical»:
visual abstracts (graphical and video-based), line 49; then you mention only «graphical abstracts» in the whole text

Please check the spelling (line 208: or – of?)

Form:
I think it is irritating, when the links to examples are embedded in the text as links (e.g. p. 6, lines 245-246, e.g. lines 314-316) and then mentioned as footnotes at the end of the text (footnotes 6, 7, 8, lines 674-676, see: Graphical abstract samples commented, line 668, there corresponding lines 679-681).

Figure 10: not yet complete?  

Author Response

(The authors gave the same response as above.)

Reviewer 3 Report

In general the coverage of the topic is good, but the the language used throughout the paper is very complex and makes the narrative of the article very hard to follow.

I would say that the methodology section needs a lot more detail. While the sources of the method are cited, the actual processes need more description for those not familiar. It would also help contextualise the results section. 

Other feedback would be on the diagrams and tables. A number of them are not well described in the body of the text. 

Author Response

(The authors gave the same response as above.)
